# Machine Learning as a Strategic Tool for Helping Cocoa Farmers in Côte D’Ivoire

**DOI:** 10.3390/s23177632

**Published:** 2023-09-03

**Authors:** Stefano Ferraris, Rosa Meo, Stefano Pinardi, Matteo Salis, Gabriele Sartor

**Affiliations:** 1Interuniversity Department of Regional and Urban Studies and Planning, Politecnico di Torino and University of Turin, 10125 Turin, Italy; stefano.ferraris@polito.it; 2Department of Computer Science, University of Turin, 10149 Turin, Italy; matteo.salis@unito.it (M.S.); gabriele.sartor@unito.it (G.S.); 3Department of Foreign Languages, Literatures and Modern Cultures, University of Turin, 10124 Turin, Italy; stefano.pinardi@unito.it

**Keywords:** cocoa farmers, low-cost smart agriculture, remote sensors monitoring, water resources forecasting, YOLO, U-NET, SITS, deforestation, drought prevision, socio-technical transition

## Abstract

Machine learning can be used for social good. The employment of artificial intelligence in smart agriculture has many benefits for the environment: it helps small farmers (at a local scale) and policymakers and cooperatives (at regional scale) to take valid and coordinated countermeasures to combat climate change. This article discusses how artificial intelligence in agriculture can help to reduce costs, especially in developing countries such as Côte d’Ivoire, employing only low-cost or open-source tools, from hardware to software and open data. We developed machine learning models for two tasks: the first is improving agricultural farming cultivation, and the second is water management. For the first task, we used deep neural networks (YOLOv5m) to detect healthy plants and pods of cocoa and damaged ones only using mobile phone images. The results confirm it is possible to distinguish well the healthy from damaged ones. For actions at a larger scale, the second task proposes the analysis of remote sensors, coming from the GRACE NASA Mission and ERA5, produced by the Copernicus climate change service. A new deep neural network architecture (CIWA-net) is proposed with a U-Net-like architecture, aiming to forecast the total water storage anomalies. The model quality is compared to a vanilla convolutional neural network.

## 1. Introduction

The Anthropocene is the geological era that began when human activities had had a global and evident effect on the ecosphere of lands, oceans, and water worldwide. Human actions are driving anthropological climate change (ACC), which is an effect of humans’ socio-economical activities.

### 1.1. The Complex System That Causes and Is Implied in Anthropological Climate Change

Climate change includes global warming: Earth’s weather patterns are caused by the emission of greenhouse gases (GHGs), carbon dioxide (CO_2_), and methane which are mostly emitted from the use of fossil fuels for energy use. Furthermore, agricultural practices and forest loss are relevant additional inducers of climate change. Twenty-three percent of GHG emissions are produced by agriculture farming, coastal economies, and the erroneous management of forests and land.

The 2019 UN *Climate Change and Land* report [1] draft by the Intergovernmental Panel on Climate Change (IPCC) states:the ACC will increase drought in some areas and extreme rainfall in other areas of the world, affecting agricultural production, the ocean economy, and the security of food supplies around the world;it will have effects on the water level by deeply involving the coastal areas and their cities;it will raise the temperature averages 1–4 degrees upward, “shifting” warm climatic zones northward, altering marine habitat and coastal economies, changing land needs and local habits, inducing tropical rains, dramatically reducing the extent of glaciers, lake levels, and the natural water reserves;it will affect the transmigration of animals and insects to areas where they were not normally present.

The IPCC 2023 poses that human activities, mainly through emissions of greenhouse gases, “have unequivocally caused global warming, with the global average surface temperature in 2011–2020 reaching 1.1 °C above the average temperature in 1850–1900” [2]. Global greenhouse gas emissions have continued to increase, with unequal historical and ongoing contributions arising from unsustainable energy use, land use and land-use change, lifestyles, and patterns of consumption and production across regions, between and within countries, and among individuals.

Heat waves, intense storms, and weather extremes are other visible effects of ACC. It affects everyone and any area on the planet; no one can consider themselves exempt, but the consequences will weigh most heavily on the weakest and most vulnerable populations and areas. Coastal zones are at risk of flooding, intense storms, and rising water levels. The Mediterranean is at high risk of desertification and weather extremes. Savage intensive agriculture in Africa is destroying the forests, with the effect of the desertification of entire regions (Côte d’Ivoire has used 90% of their forest to produce cocoa, mainly exported to western countries). This area will be subject to relevant climatic changes with severe economic consequences: wars or mass migrations are expected to affect the economy and lifestyle. Econometric models indicate that ACC has reduced global agricultural total factor productivity (TFP) by about 21% since 1961, a slowdown that is equivalent to losing seven years of productivity growth [3,4]. The effect is substantially more severe in warmer regions, such as Africa, Latin America, and Asia, where a reduction of ∼26–34% is expected [3]. The agriculture and ocean food industries have grown more vulnerable to ongoing climate change [2].

There are three categories of causes for GHG:anthropogenic activity that changes the land cover and land management;indirect effects of anthropogenic activity, such as carbon dioxide (CO_2_), fertilization, and nitrogen deposition;natural climate variability and natural disturbances (e.g., wildfires, windrow, disease).

Deforestation, scarcity of water, and other climate-related changes make cultivation more exposed. These phenomena decrease the variety of species, weaken ecosystems, and make soils more arid, pushing local farmers to compensate by using even more fertilizers and the already scarce water. These effects significantly increase GHG emissions, water dispersion, and the exploitation of territory.

### 1.2. The Motivations of the Case Study

Côte d’Ivoire is the largest producer of cocoa beans in the world, accounting for 43% of the global production [5]. However, the working conditions in the region are far from ideal, with farmers earning about EUR 1.00 per day. The combination of deforestation, inadequate rainfall, the draining of underground waters, and the impacts of climate change poses significant threats to crops and Ivorian farmers’ already fragile cocoa economy. To address this pressing issue in a developing country, it is crucial to combat climate change using methods and techniques that are affordable and accessible to the local economy.

Modeling terrestrial water content is crucial for pursuing the sustainable development goals issued by the 2030 agenda of the United Nations, especially for the sixth goal: Ensure availability and sustainable management of water and sanitation for all. Moreover, climate and land-use change could put more stress on water resources and hydro-power generation, especially if they act together [6,7,8]. Extreme hydrological events (i.e., droughts and floods) have a severe impact on a region, making environmental resource policies very relevant to the development of a country. All over the world over the last decade, nearly 1.43 billion people have been affected by droughts and 1.65 billion by floods [9]. Furthermore, global warming is going to make droughts more likely and severe [10]. Especially for Côte d’Ivoire, the likelihood of the occurrence of droughts will increase by 7.5% in the future (2051–2100) and 6.6 million people (22% of the population) will be exposed every year to these events [11]. Of course, the agricultural sector will suffer from this scenario: it implies catastrophic consequences on the already fragile economy and society of the country because agriculture is one of its driving economic sectors, accounts for 22% of the gross domestic product (GDP), about 50–70% of the total export earning and employs nearly 50% of the labor force [12].

Many studies have already analyzed the distribution of rainfall in space and time in Côte d’Ivoire and they agree on a decreasing trend, below the usual average, causing water shortage [13,14,15]. This makes Côte d’Ivoire a highly vulnerable area [13,16]. For these reasons, it is essential to develop supporting instruments for mitigation and adaptation policies. However, in many developing countries, there could be a lack of resources to build such instruments, for example, a collaborative network for monitoring surface water or groundwater [9]. In this context, open data and machine learning (ML) techniques can provide support to the sustainable management of environmental resources. For example, gravity recovery and climate experiment (GRACE) NASA mission data [17,18] provide information about the total water storage anomalies (TWSA), which could be used for determining the water shortage period, the extent of anthropogenic drought, and water resource depletion in general at a regional or catchment level [19,20,21].

Terrain sensors, satellite data analysis, neural models, and digitalization can improve the total factor productivity, by producing more with fewer inputs (less fertilizers, water, energy, capital [22]) reducing land misuse, deforestation, and GHG emissions. The research community must use the capabilities at its disposal to introduce instruments of control, methods of continuous monitoring, and techniques of forecasting that will help small farmers produce more with a reduced environmental impact. Furthermore, it is relevant to give the political decision-makers the best information and tools to control and fight climate change on a larger scale. Finally, a better perception of the green impacts of their production can induce local farmers to follow more correct habits and spread new agronomic approaches that are more ecologically attentive.

### 1.3. Technology and Machine Learning Methods at the Service of the Problem

Low-cost sensors, remote sensing, ML methods such as predictive models and more specifically deep neural networks, and cooperative approaches can enormously contribute to fighting climate change in agriculture in developing countries. The research must address the problem as a global phenomenon with multiple causes. Collaboration among experts, researchers from different areas, non-governmental organizations (NGO), and local authorities is essential to bring about lasting changes. The ability to manage territories, soil, waters, crops, and counter extreme weather also depends on how we act within communities that are part of the projects and may have limited economic resources.

The contributions of this work are manifold. On one side, Section 2.1 gives an overview of the many ingredients, low-cost or open-source, that could be used to afford the application of artificial intelligence (AI) in agriculture and spread it, especially in developing countries and companies that do not have large resources, budgets, and competences to develop them from scratch. With the intent to make this work more concrete and to demonstrate, with a proof of concept, that the proposed approach is viable, with promising and effective results, two tasks that are relevant to the agriculture economy of a developing country such as Côte d’Ivoire are selected and presented, respectively, in Section 2.3 and in Section 2.4. Moreover, a summary of the related work is presented in Section 1.4. For the first mentioned task, this manuscript illustrates a method of crop monitoring to support the identification of cocoa health status with low-cost imagery to help farmers in their production and increase their efficiency and yield. For the second task, the article shows a new model for forecasting the total water storage anomalies in Côte d’Ivoire. The goal is to identify the increase in water availability or the reduction in the dry areas at a regional scale. This can help to implement policies of intervention to prevent the phenomena of desertification and land management. We also discuss how the same methods could be useful to local policymakers and cooperatives of farmers, to identify land that is being transformed without planning or control, leading to forest depletion, deforestation, and land misuse.

Furthermore, counteracting climate change requires working concurrently at different scales, long and short. The short scale entails working at the local level, employing terrain sensors and images taken from mobile devices to help single farmers’ crop activities (see task 1). The long scale operates at the national and sub-national level and consists of using satellite data (NASA GRACE mission [17] and ERA5 Copernicus climate change service produced at the European center medium weather forecast—ECMWF [23]) to detect changes and the depletion of water resources in the terrestrial ecosystem at the 0.25° scale, harnessing and soliciting a cooperative approach among small farmers and policymakers (see task 2).

With the results, we target the local smallholder cocoa producers [24], the NGOs operating in the Abidjan area (e.g., Communauté Abel, Grand Bassam, Côte d’Ivoire [25] connected with the Gruppo Abele Foundation’s Choco+ initiative [26]) working with the local farmers and the local authorities in the context of Euro-African cooperation (cf. Pinardi et al. 2023 paragraph 3.2 [27]).

### 1.4. Related Works in Smart Agriculture and Terrain Monitoring

Cocoa plantations in Côte d’Ivoire are one of the main drivers of degradation and deforestation [28,29,30]. For this reason, tools for detecting the depletion of forestry resources are highly needed and relevant to implement policies for sustainable management. Some studies already tried to detect cocoa plantations in forest areas in Côte d’Ivoire [31,32]; and some global tools already exist to detect tree losses or tropical moist forest (TMF) degradation and deforestation [33,34]. In Africa, several factors contribute to GHG emissions. Deforestation and the expansion of agricultural activities diminish carbon absorption and increase the amount of CO_2_ in the atmosphere [35,36]. The agricultural sector emits methane, a powerful GHG [36,37], and the use of synthetic fertilizers in agriculture and waste management practices releases N2O, another relevant GHG [36,37]. Africa contributed 11% of the increase in GHG emissions since 1990 (2.3 GtCO_2_-eq) and 10% (0.7 GtCO_2_-eq) since 2010 [38]. Vast tropical rainforests and other ecosystems play a crucial role in mitigating the impacts of GHG emissions. Unfortunately, deforestation and land degradation reduce their capacity [35,36,39].

Fundamental research is essential to explore new hypotheses and gain a deeper understanding of ecological processes and their interactions [39]. Furthermore, the determination of reasonable scales and the selection of appropriate explanatory and response variables depends on understanding the context and systems under study [40].

Fortunately, some initiatives to analyze these phenomena already exist. For instance, ECMWF gives access to the reanalysis dataset of the atmospheric composition produced by the Copernicus atmosphere monitoring service: the data currently cover the period 2003–June 2022 [41] with a resolution of approximately 80 km with a sub-daily and monthly frequency. The separate global greenhouse gas reanalysis (EGG4) covers the 2003–2020 period. Another Copernicus service is ERA5. Section 2.4 proposes a data analysis of ERA5, from the Copernicus reanalysis database [23].

Today, the idea that AI can be employed in precision agriculture is widespread [42,43] following a cost–benefit analysis primarily focusing on ROI (return on investment) [44]. Little emphasis is placed on the social impacts of the processes. Instead, the concern is how the transformation can take into account the social dimensions that may influence development [27]. In this context, it is also important to address the digital costs, which are particularly relevant when working in developing countries, such as Côte d’Ivoire, where, according to the World Economic Forum [45], cocoa farmer wages average around EUR 1.00 per day. These workers contribute significantly to the national domestic product, of which cocoa production accounts for up to 20% and for 40% of the national export earnings [46].

To address the economic constraint, we have adopted a cost-effective approach by leveraging open-source software and accessing freely available data from sources such as GRACE, and ERA5 (see Section 2.4).

Taking advantage of the widespread use of mobile phones in Côte d’Ivoire, farmers can capture close-up images of cocoa plants and pods to assess the plants’ health. This method makes it easy to identify crop health problems affecting cocoa cultivation that impact up to 30–40% of its production [47,48] (see Section 2.3). The use of low-cost in-soil sensors can provide further information to search for root causes of the diseases that afflict the crop (see Section 2.1). By employing open-source methods, we can significantly reduce the financial burden. This approach ensures that even in regions with limited financial resources, it is possible to make a meaningful contribution to environmental conservation and sustainable agricultural practices.

This calls for equal attention to these constraints and opens up a new area of research related to socio-technical transition [27,49,50], where ML and sensor monitoring are not only instruments to increase yield production, or to generate new business or markets [44], but are pivotal to changing the social and economical landscape of an entire region.

With these constraints in mind, the techniques employed in this article are the following. In the mentioned task 1, the article focuses on YOLOv5m [51] for the identification of healthy/unhealthy cocoa pods (cf. Section 2.3). For task 2, the article focuses on remote sensor analysis for water resource monitoring (cf. Section 2.4) and land management, using a convolutional neural network (CNN) and fully convolutional neural network (FCN) [52] with a U-Net-like architecture [53].

## 2. Materials and Methods

This section reports on the materials and methods that are strategic for the application of ML in agriculture and come from the open-source communities of open data, open hardware, and open software. It also gives a general workflow of the activities that compose the process of knowledge discovery from data by means of ML. The following Section 2.3 and Section 2.4 describe two concrete and specific tasks, comprising the specification of the data and of the model learning process, chosen as examples of the possibilities of this approach for smart agriculture.

### 2.1. Open-Source Strategic Tools

An open-source approach to smart agriculture, and more generally big data management and stream data processing, comes with a large potential for innovation capacity thanks to the ability to freely reuse the software under open-source licenses [54]. In addition to the reuse of the products of knowledge, the presence of a large community of people and experts capable of testing and improving these knowledge products is also very important. These are factors that make the knowledge products more trustworthy, with the consequent possibility of making them explainable and reproducible.

The capability to deploy existing technology, digital platforms, and open data collections facilitates innovation by leveraging innovative services and organizational models. In particular, the evolution of open-source software communities that support software development, sharing, and reuse (such as Github [55], Joinup [56], Apache [57], GNU and the Free Software Foundation [58]) increases the diversity of potential users that act as testers and expert sources. For businesses that have less expertise in programming, open-source offers visibility into how developers manage datasets and software and helps them to cut costs because they do not need an in-house software development company. Open-source platforms, communities, and initiatives provide accessibility to a myriad of AI tools, libraries, and documentation that facilitates the development of new tools in different business sectors, using AI techniques while they are receiving feedback from experts within the community.

Open collaboration environments and open-source software (OSS) are the results of collaborative projects that speed up the reproducibility of the research (such as Wikipedia and open data repositories, such as the University of California Irvine [59] and Kaggle [60]). In turn, the presence of large volumes of data and pre-trained models has accelerated advancements in deep learning with software libraries such as TensorFlow (*r*.2.12.0) [61], DSSTNE (deep scalable sparse tensor network engine, version of the 20 February 2020) [62], and Keras (*r*.2.12.0) [63]. Other frameworks are based on high-performance computing [64,65] and cloud environments with virtual machines and software containers, such as ML-Ops (last reviewed on the 18 May 2023) [66]: they assist the developer and ML expert in the pipeline of data analysis, and provide memory space and computing power for storage and computing resources. For specific applications such as image recognition, there are web services that let users perform deep learning and even prediction without programming [67,68]. In this work, the main adopted tools were Google Colab (access date: 31 July 2023 at https://colab.research.google.com/) and Keras. The first of these is a free web-based platform, in which it is possible to implement and run Python software (e.g., *v*.3.11.5) interactively and share it with collaborators. It is possible to interact with the platform by running single lines of code or entire scripts, instantly visualizing the results. In addition, the scripts can be run exploiting CPUs or GPUs, allowing the users to efficiently train their deep learning models. Keras is the deep learning Python framework, built over TensorFlow, selected to implement the ML models presented in the rest of the article, which provides all the necessary building blocks to create neural networks and optimize them.

Finally, open data licenses such as community data license agreements (CDLA) have begun to commodify training data. These license terms will help “democratize” the overall AI marketplace by lowering the barriers to entry in the market of AI. Proprietary datasets could continue to exist, but in two versions (one of them under the CDLA license): this solution could allow everyone, including smaller players, to build credible products [69].

#### Pervasive IoT Systems

The implementation of pervasive IoT systems needs components that communicate data and perform efficient data ingestion from a stream coming from sensors. In these tasks, many open-source software projects exist that allow data ingestion, such as Apache Kafka [70], and data storage in specialized databases for time series such as InfluxDB [71].

Among the communication protocols in a distributed system, there are many protocols with different characteristics. There is message queue telemetry transport (MQTT), which is lightweight and can work in very low bandwidth networks, and hypertext transfer protocol (HTTP), and WebSocket which establish a TCP connection and are based on the request-response scheme. Constrained application protocol (CoAP) is based on a web transfer protocol and should be used with limited networks with low bandwidth and low availability. Data distribution service (DDS) adopts a publish–subscribe methodology, Advanced message queue protocol (AMQP) is TCP-based, guarantees delivery and acknowledgment, and has two levels of quality of service. The extensible messaging and presence protocol (XMPP) is based on the extensible markup language (XML). OPC unified architecture (OPC UA) is a transport-agnostic protocol and supports both request/response and publish/subscribe methods.

As regards sensors, transmitting the measured data, we refer to open sensors. Arduino sensors play a vital role in modern agriculture, enabling farmers to monitor and optimize various aspects of their crops and environment. These sensors are relatively affordable, easy to use, and can be integrated into automated systems. Here are some examples of key applications of Arduino sensors in agriculture:**Soil moisture sensors:** These sensors measure the moisture content in the soil, allowing farmers to determine the optimal time for irrigation. By ensuring the right amount of water is provided to the plants, farmers can prevent over-watering or under-watering, leading to a better crop yield and water conservation [72].**Temperature and humidity sensors:** Monitoring temperature and humidity levels is crucial for crop health. Arduino sensors can help farmers assess the environmental conditions and make adjustments accordingly, such as turning on irrigation systems or activating ventilation in greenhouses [73].**Light sensors:** Light sensors help farmers analyze the intensity of sunlight reaching the crops. This information is valuable in determining suitable planting locations, optimizing crop layouts, and even deciding the best time for harvesting [74].**Weather stations:** Arduino-based weather stations can collect data on various weather parameters such as temperature, humidity, wind speed, and precipitation. Farmers can use these data to anticipate weather changes and prepare for potential adverse conditions [75].**Crop health monitoring:** Sensors such as pH sensors and nutrient level sensors can provide insights into the health of the crops and soil. Farmers can adjust fertilization and nutrient application based on real-time data, leading to healthier plants and better yields [76].**Pest detection:** Some Arduino sensors can identify pests and diseases early on by detecting specific patterns or changes in the environment caused by these issues. This helps farmers implement targeted pest control measures, reducing the need for excessive pesticide use [77].**Automated irrigation systems:** By integrating Arduino sensors with irrigation systems, farmers can create automated setups that respond to real-time data. These systems can turn on or off the irrigation based on soil moisture levels, weather conditions, and crop requirements [78].**Crop growth monitoring:** Sensors such as ultrasonic distance sensors or infrared sensors can measure crop height and growth rate. This information allows farmers to track the development of their crops and make timely decisions regarding pruning or harvesting [76].**Livestock monitoring:** In addition to crop-related applications, Arduino sensors can also be used to monitor the health and behavior of livestock. For example, sensors can track the body temperature of animals, detect estrus in cattle, or monitor feeding and drinking habits [79].**Automated greenhouse systems:** Arduino sensors can be integrated into smart greenhouse systems, controlling temperature, humidity, and ventilation automatically to create an optimal environment for plant growth [80].

Overall, Arduino sensors offer an affordable and accessible way for farmers to gather valuable data, optimize their farming practices, and make informed decisions to enhance productivity and sustainability in agriculture.

### 2.2. The Pipeline of Knowledge Discovery from Data

Figure 1 shows the pipeline of activities for achieving knowledge discovery from data, composed of multiple steps, from the collection of data that are the measured variables coming from different sources and sensors to the final activities of ML.

The data sources might be, at the large scale, satellites that collect images of the territories; at a lower scale, drones or personal devices such as mobile phones or tablets; and at the small scale, specific sensors installed in the territory monitoring physical quantities or chemicals. The collected data (step 1) are transferred by means of the data transfer protocols to be stored in files or databases on remote servers (step 2). The obtained datasets could be shared, published, and reused so that the effort and costs for the collection of data are cut and amortized. If this happens, the previous two steps do not need to be executed at any time, and the pipeline starts from the following steps. Data merge (step 3) is needed for a complete and profitable data analysis and ML model training, in which sensor data are fused, and consolidated at the same level of (spatial and temporal) granularity, and datasets are integrated. The following steps encompass the proper data analysis activities. In step 4, there are possible data transformation phases, that could prepare new features from the original ones, clean data for the missing or noisy values, discard the subsets of features that are of little use, etc. Some ML methods, such as deep neural networks, support vector machines, principal component analysis, singular vector decomposition, ensemble methods, and decision trees include some operations of feature selection or transformation. In these cases, the pipeline could omit step 4 and proceed to the following step 5, specifically focused on the activities of ML: model training, testing, validation, and application with the final goal of obtaining a digital model of a physical phenomenon or of an interesting target variable, whose estimation is based and reconstructed based on observed and measured physical quantities.

### 2.3. Task 1: Cocoa Pods Classification Model

To guarantee good quality beans, cocoa plants must be continuously monitored from ripening to harvesting. The accurate selection of cocoa beans is a crucial undertaking that significantly impacts the subsequent activities and, consequently, the final product quality. Recognizing ripeness and the absence of anomalies in the beans and pods is still a manual activity and assumes the presence of expert operators employed in the field. Phytophthora and Moniliophthora are the main causal agents among cocoa diseases, causing up to 38% of the global cocoa harvest loss [46,47,81]. Some species of these agents are present worldwide. In contrast, others are endemic to specific regions (e.g., *Phytophthora megakarya* in West Africa, *Moniliophthora perniciosa* in Latin America and the Caribbean, etc.). The diseases caused by Phytophthora manifest themselves in the form of black pod rot, stem canker, leaf, and nursery blights. Instead, witches’ broom disease and frosty pod rot characterize the presence of Moniliophthora [81]. Therefore, detecting diseases and blocking their spread is a critical activity.

Recently, to provide partial support to operators, some AI tools were proposed [82,83,84,85]. This article suggests a possible approach to support farmers in recognizing good-quality cocoa beans using state-of-the-art AI tools, such as the deep neural network architecture YOLO [51]. With open-source tools, it is possible to create a model to detect specific diseases without significant monetary or computational resources.

This task needs the execution of steps 1, 4, 5 of the pipeline in Figure 1. First, the dataset, composed of a collection of images, was downloaded and it was almost ready to be employed in training (step 5). However, the dataset needed a slight modification (executed as step 4 of the pipeline explained in Section 2.3.1) before it was passed as input to the training phase of the ML model (step 5).

#### 2.3.1. Data

Task 1 employed an open-source labeled dataset (containing information from Kaggle https://www.kaggle.com/datasets/serranosebas/enfermedades-cacao-yolov4 (accessed on 19 May 2023), made available here under the Open Database License (ODbL)) identifying healthy and damaged beans affected by Moniliophthora and Phytophthora diseases, in three different classes (respectively, called Healthy, Monilla, and Fito). More specifically, the dataset contains 312 images of dimensions 3120×4160: 107 are from the Fito class, 105 from Monilla, and 100 from Healthy. As previously stated, Moniliophthora and Phytophthora agents are the most dangerous, and detecting them in time can save a significant part of the harvest. These data were already labeled and almost ready to use, but it was necessary to adopt the following specific structure of the dataset:



Inside the dataset folder, the dataset.yamlfile contains the dictionary of the labels (0: Fito, 1: Monilla, 2: Healthy) and the path of the dataset for training and testing. In addition, each image has a textual file containing the classes’ bounding boxes contained in the image (e.g., the file labels/image1.txtcontains the labels for the image images/image1.jpg). Each line of a “txt” file has the form <class x_center y_center width height>, defining the number of the class, and the center position, width, and height of each bounding box.

#### 2.3.2. Model

For task 1, the pre-trained YOLOv5m model (https://docs.ultralytics.com/yolov5/ (accessed on 19 May 2023)) was used, which is an improvement of the original YOLO model [51]. It was trained for the last layer of the model on the cocoa diseases dataset using Google Colab (https://colab.research.google.com/ (accessed on 19 May 2023)) Previous systems such as region-based CNN (R-CNN) first generate potential bounding boxes and then run a classifier for detecting objects inside them. However, post-processing is required to eliminate duplicates and produce a valuable output. The YOLO approach is different: it uses a unique CNN that simultaneously predicts different bounding boxes and their associated probabilities. YOLO does not look at inputs locally but globally, using information from the entire image for each simultaneous box prediction; not incidentally, YOLO stands for “you look only once”. This structure makes YOLO so efficient and effective that it can be used for object detection in smartphone applications [86,87,88].

In addition, it is worth noting that this version of YOLO integrated different types of data augmentation to make its training more robust and avoid overfitting. These techniques included mosaic augmentation, copy–paste augmentation, random affine transformations, mixUp augmentation, albumentations, HSV augmentation, and random horizontal flips.

The model was trained on the training set for 50 epochs using a batch size of 16 images. The model was trained using Google Colab, exploiting its NVIDIA A100-SXM4 40 GB GPU, completing 50 epochs in 47 min; (the script for training YOLO5m model is publicly available at https://github.com/gabrielesartor/cocoa-pods-diseases-detection (accessed on 19 May 2023)). Figure 2 shows the loss reduction on the training set with the number of epochs.

### 2.4. Task 2: GRACE Prediction Model

Previous studies have already shown the effectiveness of using deep learning for remote sensing data, in particular for satellite image time series (SITS) and GRACE data [89,90,91]. However, specific application studies for Côte d’Ivoire are lacking.

The aim of this task is to develop an instrument to monitor and predict the anomalies of water resources. In particular, the objective is to model TWSA, predicting the value of the next month (at time t+1) considering the values of meteorological and land variables at the previous month (time *t*). A *vanilla* CNN was considered as a baseline. A fully convolutional neural network (FCN) [52] with a U-Net-like architecture [53], named CIWA-net (Côte d’Ivoire water anomalies network) was implemented and compared with the baseline. These architectures were selected for the neural networks because previous studies already reported their good performance in modeling spatio-temporal phenomena and SITS [92,93,94,95]. Moreover, FCNs are more suited for pixel-per-pixel tasks [53,96]. In this work, these two different architectures—vanilla CNN and CIWA-net—were tested in different implementations. In each implementation, a different autoregressive component was integrated, working as an additional input, that consisted of the delayed target variable TWSA, at a different time lag. In this way, we tested if the vanilla CNN or CIWA-net could enhance their performance by integrating the temporal information of the target without a deep modification of their architecture.

#### 2.4.1. Data

This research employed GRACE mascon solution from https://www2.csr.utexas.edu/grace/ (accessed on 19 May 2023) [97] monthly data. The GRACE satellite measured the variation in the Earth’s gravitational field for each month and then estimated the changes in the equivalent water thickness (EWT) at a spatial resolution of 0.25° (25 km). EWT is related to the total amount of water stored and available for a unit volume. However, the native cell grid resolution of the GRACE data is larger than 25 km—about 120 km—because the Earth’s gravitational field estimations vary slowly in space. For this reason, it is advised to be careful in interpreting the GRACE data in basins smaller than 200,000 km^2^. Notwithstanding these limitations, the aim of the presented task 2 was to build a model able to predict GRACE images that represent the target variable in a spatial map. Therefore, one should not attribute these limits, due to the low resolution of the input images, to the neural network models used for prediction.

The changes, or anomalies in EWT, were calculated with respect to the 2004–2009 mean baseline and they represented the total terrestrial water storage anomalies (TWSA) from soil, snow, surface water, groundwater, and aquifers. The GRACE mission started in April 2002 and ended in June 2017, but in March 2018, the new GRACE-Follow-on (GRACE-FO) mission started. Given the missing values from October 2017 to March 2018, we obtained monthly data from April 2002 to June 2017. Given the relevance of the temporal relationship of the phenomenon, the period from April 2002 to April 2015 was considered as the training set, roughly 85% of the available observations, and the test set as the period from May 2015 to June 2017, nearly 15% of the available observations.

Meteorological and land data from the fifth-generation ECMWF reanalysis for the global climate and weather European dataset (ERA5) [23] were used. ERA5 contains a large number of atmospheric, land, and oceanic climate variables, combining model data with observations from the year 1940 up to the present time. The spatial resolution of the data was 0.25° with an hourly frequency. However, given the time resolution (monthly) of the dependent variable, monthly average ERA5 data were adopted. Furthermore, the same time domain that was used for the GRACE data (i.e., from April 2002 to June 2017) was employed and the data for training and testing were split accordingly. Among all the ERA5 variables, 10 features that were likely to be associated with the target variable (listed in Table 1) were selected. All these features were considered as the different channels of an image available at a time *t*. Hence, for each time step, we had an ERA5 image made up of 10 channels. Referring to the pipeline in Figure 1, task 2 started from step 3 because the data were open and freely available.

#### 2.4.2. Data Preprocessing

Focusing on Côte d’Ivoire, both GRACE and ERA5 were cropped to a square with coordinates −8.875° and −2.125° of longitude, and 4.125° and 10.88° of latitude. Given the difference between the ERA5 and GRACE reference grids, the ERA5 data were reprojected onto the GRACE reference grid.Thanks to this projection operation, the 28 × 28 resolution images in ERA5 and GRACE had corresponding pixels referencing the same geographical coordinates (obtained in step 3 of the pipeline). More precisely, for the target variable, for each month in the training or test set, a 28 × 28 resolution image was present: each pixel value in an image had a single TWSA value for that specific month in the geographical area covered by the pixel. Differently for the input features coming from ERA5, for each month, a 28 × 28 resolution image existed but with ten values associated with each pixel (i.e., ten channels representing the ten selected input features from ERA5 listed in Table 1). Therefore, each feature value in the input referred to a specific month and a specific cell in the geographical area covered by the pixel.

The GRACE data exhibit some monthly missing values (NA) in the selected time window due to technical reasons [98]. For filling the missing monthly data (step 4 of the pipeline) a linear interpolation was adopted over the time dimension. In other words, for each month $t\in T_{missing}$ without observations, a new sample $y_{t}$ was created by linearly interpolating the samples $y_{t-1}$ and $y_{t+1}$ (where *y* denotes an entire image).

Given the different ranges in variations in the variables (see the column range in Table 1), the input and output were normalized (step 4 of the pipeline). More precisely, each feature of the input was standardized such that $\frac{x_{f}-\mu_{f}}{\sigma_{f}}$, where f∈F represents a specific feature, and the mean and the variance ($\mu_{f}$, $\sigma_{f}$) were calculated considering the pixels of all the images in the entire observed period. Finally, the output was scaled between 0 and 1 using the *min–max* feature scaling formula $\frac{Y-Y_{min}}{Y_{max}-Y_{min}}$.

#### 2.4.3. Model

As already mentioned, task 2 implemented and compared two different models, a vanilla CNN, and CIWA-net using Google Colab (step 5 of the pipeline). The best model was adopted to prove the feasibility of the task goal. The mean absolute error loss was used for both architectures, and the ReLU function was used for the activation of every layer. Every convolutional layer adopted zero-padding and a stride equal to one (zero-padding is used to fill the pixels in the contours to make the input data of the same size as the output; the stride is the unit shift between one window and the next one), while a stride equal to two was used for downsampling the images of the CIWA-net model because it needed to reduce the size of the input. Transposed convolutions were performed for the reverse operation in order to obtain back images of the original size. Therefore, by upsampling, the image resolution was augmented again and the original input size was restored, thanks also to the skip connections present in the CIWA-net model. The implemented architectures are depicted in Figure 3 and Figure 4 (the reader can download the script for their creation and training at the public repository: https://github.com/Matteo-Salis/CIWA-net (accessed on 31 July 2023)). Both models took as input images of dimensions (28,28,n_features). As previously explained, 28 × 28 are the dimensions obtained after preprocessing the GRACE and ERA5 images, and n_features is the number of features selected for the experiments and it is also the number of channels. The output of the obtained neural network models is an image of dimensions (28,28,1), because for each pixel of the image only one feature is estimated, the target variable. To integrate an “autoregressive” component, different scenarios were developed:n_features=10, when models took in input of only the 10 features of ERA5, listed in Table 1 at time *t*, i.e., an image with 10 channels;n_features=10+δ, where δ stands for the number of additional channels, each of them made by a delayed GRACE data image. Hence, for δ=2 the input image had 12 channels, 10 of which were ERA5 variables at time *t*, one was GRACE data at time *t*, and the last was GRACE data at time t−1, all trying to predict GRACE at time t+1.

## 3. Results

This section shows the results produced by the two tasks outlined above, the cocoa pods classification (task 1), and the CIWA-net prediction model (task 2).

### 3.1. Task 1: Results

The results of the YOLOv5m model estimations are shown in Figure 5 through the confusion matrix on the test set with leave-one-out cross-validation.

It represents the performance of the implemented models in discriminating different classes. The model detects almost every time (98%) a healthy fruit, while it has some difficulties in differentiating between diseases. A more complex model could be implemented to discriminate better. However, given the aim of this task is to prove the feasibility of an instrument that should be used on smartphones, it could be worth giving up a little bit of performance in favor of a faster and lighter model. Furthermore, detecting a specific disease can also be difficult for experts. A free, fast, and open tool that could tell farmers if fruits are healthy or not could be a tremendous help for helping in crop monitoring and yield forecasting.

Some predictions for a validation set, extracted randomly, are shown in Figure 6.

As the confusion matrix revealed, the trained model detected healthy fruits very accurately, while it confused Monilla and Fito classes around 30% of the time. When the model was more confused, and the chance of making a mistake grew, the probability associated with each predicted bounding box and its class was lower (such as in the first images on the left in Figure 6). This should be considered as a warning and in these cases more detailed analyses should be carried out on the fruit. At the first report of a disease, the first trivial practice is to intervene to sanitize the cocoa plant and to remove the damaged fruits. Consequently, the presented YOLOv5m model could be extremely useful for acting immediately on-site when the first symptoms occur on fruits, and warning experts in case of extended and unclear fruit conditions. When the infestation is particularly extended or the disease is likely to damage the whole plant, the farmer can be helped by experts to avoid the spreading of the threat. For instance, Moniliophthora can be eradicated by simply sanitizing the plants and surroundings by removing infected pods and brooms, or by inserting natural enemies of the agent (e.g., specific invertebrates and mycoparasites). Instead, for Phytophthora, one can intervene with phytosanitary (every four days) measures, improving soil drainage, since it proliferates in wet areas, inserting plastic covers to prevent insects from carrying the disease elsewhere, or using sprays for disease prevention [81]. At the moment, Moniliophthora is not present in Africa, but some studies claim that Western Africa presents an environment suitable for the proliferation of these diseases as well, as the wet environments in which the pathogen proliferates are also present in West Africa [99]. Therefore, it is important to avoid importing fruits attacked by this agent to prevent its spreading, causing dangerous effects on the environment.

Thus, given the fact that an ML model is able to generalize from specific examples to novel examples, not previously seen before, it could be possible to apply task 1’s solution in all the different areas of the world where cocoa is grown. In fact, Moniliophthora and Phytophthora cause the most common and destructive diseases in cocoa cultivation, and there are many variations worldwide. Therefore, task 1’s solution should be more extensively used and this is easily achieved by spreading this model via the open-source communities.

### 3.2. Task 2: Results

In relation to the results of the CIWA-net prediction model, i.e., task 2, Table 2 shows the mean squared error (MSE) and mean absolute error (MAE) for different scenarios of the inputs in the training set and in the test set, in which for a scenario we set a specific δ value of the autoregressive term. The errors are used to compare the vanilla CNN and the CIWA-net models.

CNN models seemed to overfit the data: in fact, they achieved far fewer errors in the training set but similar or higher errors on the test set than CIWA-net. For δ=0, the CNN model performed better than CIWA-net in terms of the test set, while for δ>0, the CIWA-net model seemed to take more advantage of the additional input represented by the GRACE delayed channels, and succeeded in achieving smaller test errors (except for δ=4). For δ=1, the CNN model outperformed CIWA-net only if one considers MAE in the test set. However, the CNN model had an MSE in the test set higher than the respective errors of CIWA-net: this meant that CIWA-net penalized strongly higher errors.

Apart from the slightly better performance among neural networks with δ>0, it is evident that introducing one or more (δ>0) GRACE delays, given as additional input channels for both CNN and CIWA-net, allowed them to enhance their performance. Even if the differences were quite small (10${}^{-3}$ order of magnitude), introducing too many GRACE delays appeared not to be worth it because it increased the test errors or did not significantly improve performance. The optimal number of delays appeared to be one or two. Maybe this could be due to the fact that the input images had already a high number of channels (ten) and adding too many channels seemed to condense too much information in a single image. The best results were obtained using δ=1 for the CNN model, and δ=2 for the CIWA-net model, the latter achieved the overall best scores (in underlined bold in Table 2) and it was selected as the final model for task 2. Hence, RMSE and the normalized root mean squared error (NRMSE) obtained by normalizing the error for the range of the data in the test set (from May 2015 to June 2017) were computed as in Equations (Equation 1) and (Equation 2).
(1)$$\begin{array}{c} RMSE=\frac{1}{N}\sum_{i=1}^{N}{\left(Y_{i}-Y_{i}^{*}\right)}^{2} \end{array}$$
where $Y_{i}$ represents the true value and $Y_{i}^{*}$ represents the corresponding predicted value.
(2)$$\begin{array}{c} NRMSE=\frac{RMSE}{Y_{max}-Y_{min}} \end{array}$$
where $Y_{max}$ and $Y_{min}$ are, respectively, the maximum and minimum true values.

Figure 7a,b show 28 × 28 images with Côte d’Ivoire’s borders, in which for each pixel its corresponding color is related to the model RMSE and NRMSE on the test set for that pixel.

It is evident that in the southeastern part, the model made more errors. This could be due to the fact that in this area Eastern Guinean forests are present and this is a completely different environment scenario from the middle and northern savannah region. Furthermore, the GRACE data (that contained the target variable to be predicted) exhibited the highest variance in this area, while the ERA5 data (that contained the input features to the prediction model) had the lowest variance. This made the task of explaining the TWSA variability using the ERA5 data more difficult in this area. The highest NRMSE values are in the southern region, where the sea is present, and this is due to the very low variability in the GRACE data over the sea. It is worth noting the highest errors are outside the region of interest (Côte d’Ivoire). Given the overall standard deviation of 9.4 cm for all the available GRACE data from April 2002 to June 2017, the RMSE values inside Côte d’Ivoire (in which the maximum RMSE was 6.15 cm) could be regarded as acceptable, even if model improvements are needed. The maximum NRMSE value inside Côte d’Ivoire’s borders is 31.6%, which is a fairly good result and it makes the model a good instrument for supporting policy implementations.

To summarize how the model performs in time, Figure 8 shows the mean of the TWSA true and predicted values, RMSE, and NRMSE. The mean TWSA was computed by averaging all the values (i.e., pixel values) for the corresponding month. In this case, RMSE was computed on the errors between true and predicted values for all the pixels of the respective month; NRMSE is the RMSE normalized with respect to the range of the true data (i.e., pixel values) of the corresponding month.

The mean TWSA values show seasonality induced by the alternation of the dry and wet seasons in the country. This behavior is correctly detected by CIWA-net. The RMSE series shows a peak for the date 1 December 2016 that could be due to a very different spatial distribution of the target in the two consecutive months (see Figure 9a). Similar explanations could be given for the two other peaks (1 October 2015 and 1 September 2016). The NRMSE in Figure 8 exhibits a similar behavior and its maximum value is 34.9%, a good result considering the small size of the dataset.

Figure 9a,b show some predictions made by the selected model in task 2 for some dates of the test set (from May 2015 to June 2017), including the one for which the model achieves the highest RMSE (1 December 2016) and some of the best performance model dates (1 October 2015 and 1 December 2016). In general, CIWA-net was able to correctly predict the spatial and temporal evolution of TWSA, apart from some anomalous and abrupt changes (e.g., on 1 December 2016) with respect to the previous month, for which the model needs to be improved. The difference between the resolution of ERA5 and the native one in GRACE caused the model outputs to vary more than the true ones. In fact, the true GRACE values remained constant in the neighboring pixels, which was not the case in the ERA5 data. However, this is not a relevant problem for the final aim of the present research: the model should be used by policymakers for a national or sub-national forecast of the major TWSA, and not necessarily at a resolution of 0.25°. Considering the above-mentioned resolution limitations, for this task CIWA-net succeeded: it is able to detect the major changes, and their approximate locations in space and time.

## 4. Discussion

The first part of this article (Section 2.1) shows that different types of low-cost sensors and open data can be employed to help developing countries manage their resources. On one hand, local farmers can be supported by tools for on-site decision-making or monitoring aspects (described with the example in task 1) and, on the other hand, policymakers can have a regional-level perspective analyzing satellite data (described with task 2). The two tasks are further discussed in the following.

### 4.1. Task 1

As demonstrated in Section 2.3, with a relatively simple dataset for classification purposes, it is possible to build classification apps to support local farmers in their activities. Nowadays, local farmers endowed with a smartphone could be able to collect on-site pictures of their crops and have them labeled by an expert. With such a dataset, it is feasible to create ML classifiers that can help farmers identify particular features, which inexpert workers could not detect. Locals can be instructed on how to correctly construct classification datasets for training ML models. Note that using a free GPU usage service (e.g., Google Colab), the YOLOv5m model could be used as a “pre-trained” ML model that can be easily adapted to other similar tasks, such as recognizing fruit ripeness [82]. Finally, a model implemented using libraries such as TensorFlow, Keras, or PyTorch can be easily converted into a format running on a smartphone, making it usable by the local farmers who can benefit from their activities.

The accuracy of the presented model based on YOLOv5m image segmentation, as depicted in Figure 5, is 98% for detecting healthy fruits, while the other two classes are often confused (around 30% of the time). However, if the aim of the farmer is just to detect healthy and damaged fruits, the tool can be considered accurate. YOLOv5m is not the best model for object detection in terms of accuracy, but it is revealed to be particularly suitable for real-time tasks, as in our case. Furthermore, the model has been trained over a small dataset, given the scarcity of this kind of data (only 312 images). Nevertheless, even though the dataset needs to be extended, the result should be considered interesting since the model internally implements a data augmentation step that makes it more general and robust with new images.

Despite some recognition and classification work on cocoa presented in other articles, all of them presented different datasets and, consequently, the different results were hard to compare. Bastidas-Alva et al. used YOLOv5 for the recognition of cocoa ripening, presenting an accuracy of the model of 60.2%, lower than the results presented in this article [82]. However, the use cases are slightly different, and probably, the lower performance could be caused by a task that perhaps is harder due to the higher similarity in some cases between ripe and unripe fruits. Other works demonstrated the effectiveness of YOLO in object classification tasks, achieving good results in detecting ripeness, for instance, of tomatoes [83], and blueberries [84]. Instead, Harvyanti et al. addressed the applicability of image classification to detect the vascular streak dieback (VSD) in Indonesian cocoa. Classifying damaged and undamaged single leaves they were able to reach 98% accuracy using other models [100] (Darknet-19, AlexNet, and SqueezeNet). However, that task was focused on single-leaf image classification (healthy or VSD), while this article took into consideration a task of object recognition with three labels, which is significantly different.

Baba et al. developed a similar application for distinguishing healthy and cocoa fruits damaged by diseases and pests [85]. The application was tested on different smartphones showing results for the accuracy higher than 80%. However, the implementation details are missing and it is not possible to compare the employed architectures.

The employment of YOLO, as used in this work, instead of Faster R-CNN, as used in these previous works, means achieving faster responses and slightly sacrificing prediction accuracy, which should be preferable for real-time applications. It is worth noting that promising results could be reached using newer versions of YOLO or other learning algorithms, such as single-shot detection (SSD) [101].

### 4.2. Task 2

Section 2.4 demonstrates the possibility of using the GRACE and ERA5 datasets to build a model based on deep neural networks of prediction for drought a month in advance. Similar studies for such a small hydrological basin using GRACE data are lacking in the literature. Most research focuses on bigger areas or countries [90,91,102]. Having larger areas of interest makes it possible to create a larger training dataset, and increases the performance of the estimated models, with the downside that data collection, elaboration, and model training times increase as well. Therefore, there is a trade-off between building a general model for predicting GRACE in large and different areas, and a model for a specific geographic area (e.g., Côte d’Ivoire) in which a smaller dataset could be sufficient. In the studies presented in [90,91,102], some statistical models such as SARIMAX, multiple linear regression, and some deep learning models such as nonlinear autoregressive with exogenous input (NARX), CNN, GAN, and deep convolutional autoencoders are used to model GRACE data. In the above-cited studies, apart from the meteorological data, different inputs were used: for example, the normalized difference vegetation index (NDVI) and the delayed value of the features in the NARX model. As a general consideration, deep learning models outperformed statistical ones, and this could be due to the complexity of the phenomenon under study. Different error metrics and normalization for the RMSE were employed in the literature; here, RMSE and NRMSE, as specified in Equations (Equation 1) and (Equation 2), were adopted because of the proximity to zero of the target mean and variance in some months. Given the different evaluation metrics and because other studies refer to other hydrological basins, a strict performance comparison to other research would not be sensible, especially considering the complexity of water storage and flow phenomena in different climatic areas. Notwithstanding, our CIWA-net shows performance in line with that reported in [90,91,102]. CIWA-net produces information that could represent a decisive improvement for developing countries and would allow them to manage water resources in the short–mid-term. This proves the feasibility of building such a tool to support policymakers or local authorities for resource management, climate change mitigation and adaptation, and in general environmental policies where water resources trend estimations are needed.

As regards the choice to start from the GRACE dataset as a source of data in the developed pipeline of task 2, this choice depended on the lack of ground data in this area of the world (and this occurs in other developing countries). However, the GRACE dataset is quite limited by its low spatial resolution, which is approximately 25 km/pixel and already extrapolated from an original granularity of 120 km (as explained in Section 2.4.1). Another issue that is worth considering is the short time period covered by the GRACE data (only after April 2002) that forbids models to learn long-term cyclical behavior and relations between the target and the features. Therefore, this dataset needs to be improved significantly in the future in order to detect smaller basins and bigger ones more accurately. Anyway, over the years, satellite missions have increased the acquisition frequency and the ground resolution of their products. This will contribute to building more precise predictive models from a temporal and spatial resolution perspective. Additionally, when it is feasible, this information could be improved by integrating data acquired by sensors in the ground, resorting to the so-called downscaling methods (to increase the spatial resolution by any procedure that infers high-resolution information from low-resolution variables).

In summary, our article shows the potential of cost-effective sensors and freely available satellite data to empower developing countries in managing their resources effectively. By providing tools and models accessible to both policymakers and local farmers, we can make significant strides in resource management and environmental monitoring without relying on expensive instruments.

### 4.3. Future Work

The system developed for task 1 could be implemented as a smartphone application and extended with the aim of discriminating among a higher number of diseases. This would require integration into the dataset of more disease examples, with the help of experts, bringing domain knowledge.

Developing countries will benefit the most from task 2, with low-cost instruments to tackle climate change. One element that will improve CIWA-net’s performance is the training set dimension. Future analyses could also take into consideration similar countries. Furthermore, it would be worth retrieving more features (e.g., NDVI and delayed values of the inputs variable) as in [90,91,102], and some proxy data for anthropogenic pressure on water resources. Other types of CNN-like architectures could be well suited for this task, for example, tempCNN, TCN, convLSTM, convolutional autoencoder, and GCN [90,93,103,104,105].

It is possible to implement models for detecting degradation and deforestation using satellite images from the Sentinel-1 and Sentinel-2 European Space Agency (ESA) missions. Sentinel-1 appears relevant for forest change detection, especially to overcome the problem of Sentinel-2 on cloud coverage and adverse meteorological conditions [106,107,108,109]. The CIWA-net model implemented in task 2 in Section 2.4 could indeed be restructured for forest monitoring, producing output of a classification map detecting deforestation and degradation.

## 5. Conclusions

The present study focuses on the urgent need for climate change mitigation and adaptation in developing countries such as Côte d’Ivoire, where the economy heavily depends on agriculture, and farmers face challenges due to climate change. By using cost-effective AI models and open-source software, the potential to provide valuable support to farmers and policymakers is discussed, aiding in sustainable land and water management practices. The study emphasizes the importance of collaboration among researchers, farmers, NGOs, and policymakers to bring about lasting changes working at different scales.

The article discusses the use of deep neural networks (YOLOv5m) to distinguish healthy cocoa plants and pods from unhealthy ones using mobile phone images, thus helping local farmers improve cocoa production at a low cost. Additionally, the article shows a method of TWSA forecasting using fully convolutional neural networks (FCN) with a U-Net-like architecture, called CIWA-net.

The new architecture, CIWA-net, is discussed and shown through comparison with a vanilla CNN model. The YOLOv5m accuracy is shown using a confusion matrix.

The proposed models for cocoa pod classification and water resources forecasting show promising results, with the potential to help farmers, NGOs, and local authorities to make informed decisions to address climate change impacts and improve agricultural practices.

## Figures and Tables

**Figure 1 sensors-23-07632-f001:**
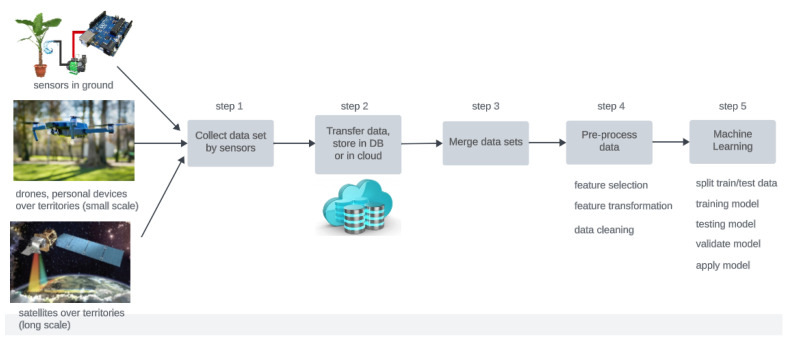
A general pipeline of activities for the collection, elaboration of data, and production of ML models.

**Figure 2 sensors-23-07632-f002:**
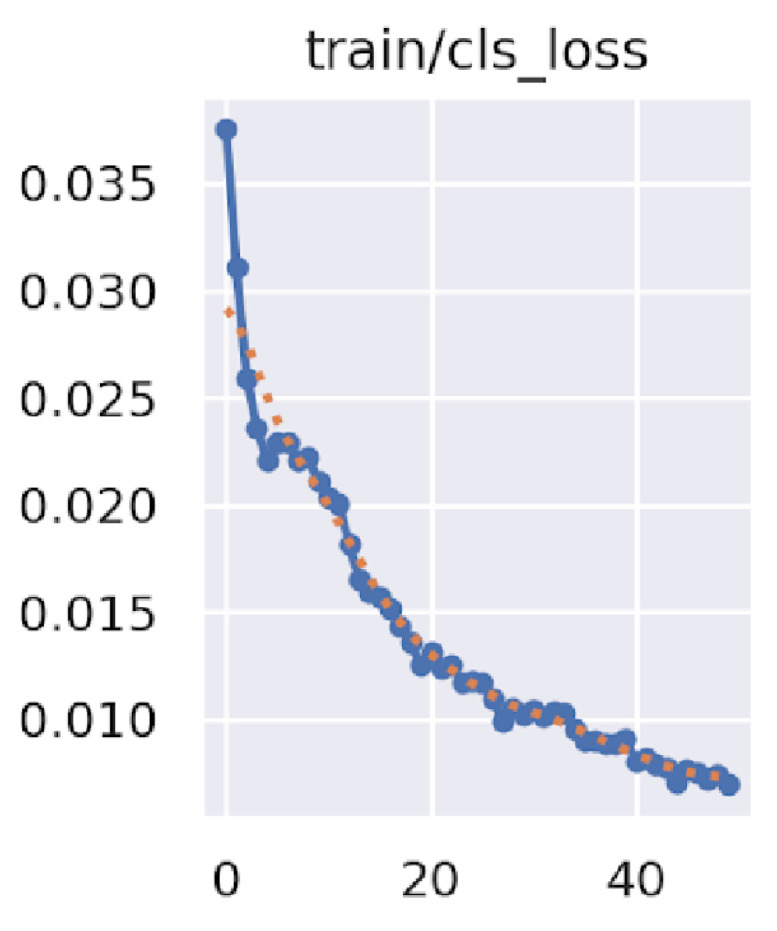
The loss over the training set for each epoch (red dots are an interpolation).

**Figure 3 sensors-23-07632-f003:**
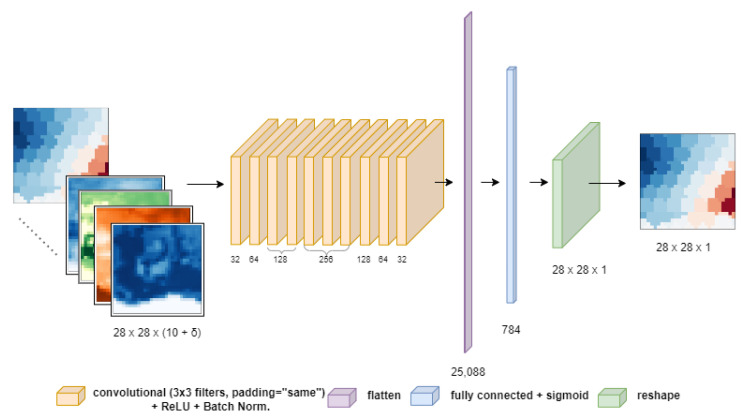
Architecture of the ‘vanilla’ CNN. The numbers under each layer represent the number of filters. The numbers under the images or the reshape layers represent their dimensions.

**Figure 4 sensors-23-07632-f004:**
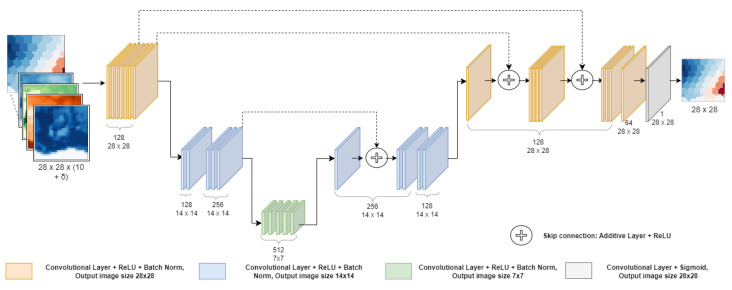
Architecture of CIWA-net. The numbers under each layer represent the number of filters and the output image resolutions; curly brackets are used for layers with the same numbers.

**Figure 5 sensors-23-07632-f005:**
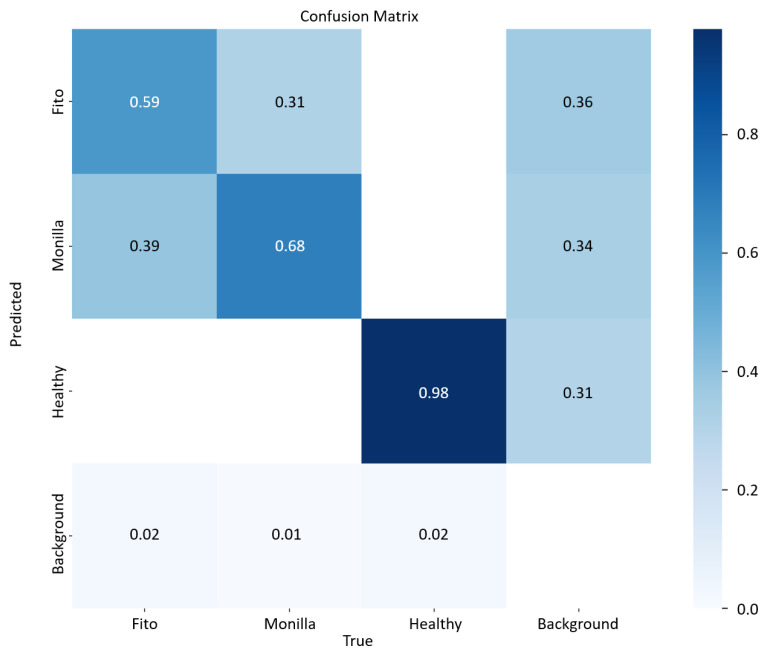
Confusion matrix of the YOLOv5m model trained on the cocoa diseases dataset.

**Figure 6 sensors-23-07632-f006:**
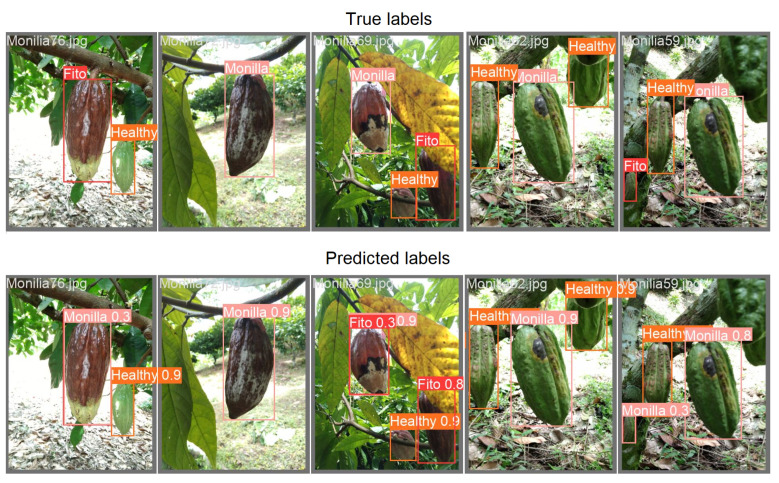
Examples of labeled images in the training set and the corresponding prediction with probabilities.

**Figure 7 sensors-23-07632-f007:**
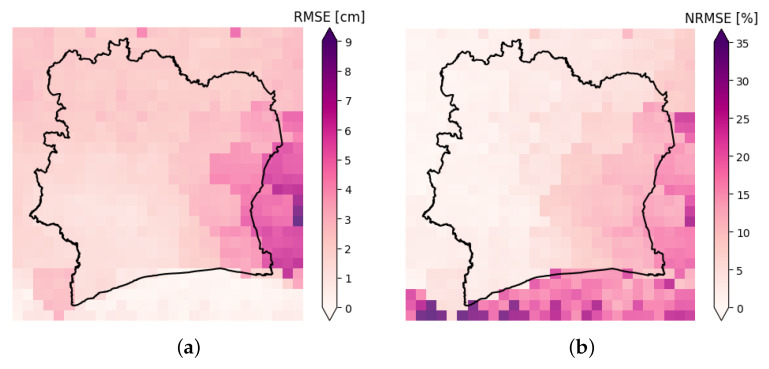
(**a**) Root mean squared error (RMSE) for each 0.25° pixel computed on the test set (from May 2015 to June 2017). (**b**) Normalized root mean squared error (NRMSE) for each 0.25° pixel computed on the test set (from May 2015 to June 2017) (NRMSE is the RMSE normalized with respect to the range of the true data for each pixel). For both (**a**,**b**) errors are computed using CIWA-net with δ=2 predictions with respect to the true TWSA GRACE values in cm.

**Figure 8 sensors-23-07632-f008:**
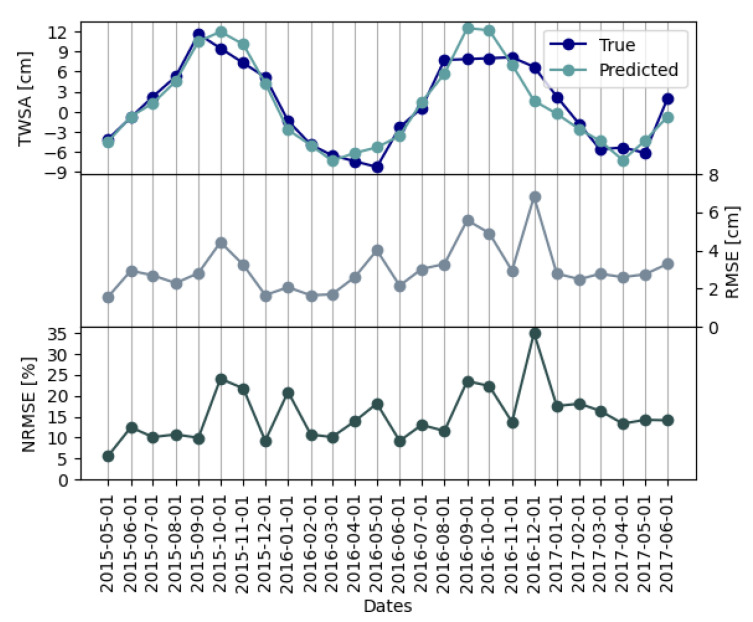
True and predicted mean TWSA time series for the test set (from May 2015 to June 2017). Mean TWSA was computed by averaging all pixel values of the corresponding month. The RMSE time series was computed considering the errors of all the pixels for the respective month on the test set. The NRMSE time series was computed by normalizing the RMSE with respect to the range of the true data for the corresponding month on the test set. Errors were computed using CIWA-net with δ=2 predictions with respect to the true TWSA GRACE values in cm.

**Figure 9 sensors-23-07632-f009:**
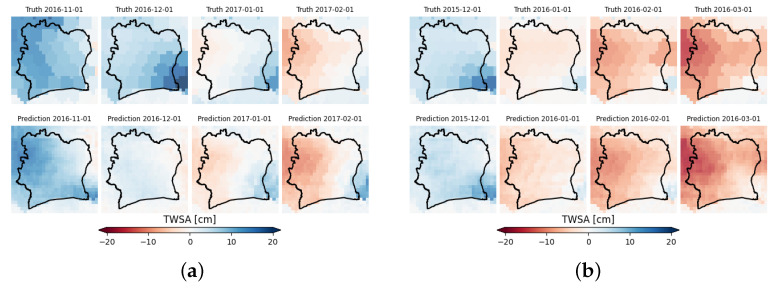
Ground truth (first row) and prediction (second row) of GRACE TWSA from 1 November 2016 to 1 February 2017 (**a**), and from 1 December 2015 to 1 March 2016 (**b**). Predictions were made using the CIWA-net model with δ=2. In (**a**), 1 December 2016 is the date for which CIWA-net performed the worst. In (**b**), 1 December 2015 and 1 March 2016 are the dates for which CIWA-net performed the best.

**Table 1 sensors-23-07632-t001:** ERA5 features selected for the neural network inputs with their ranges of variation and units of measure.

Feature	Range	Unit
Surface net solar radiation	1.319 × 10${}^{7}$	J·m${}^{2}$
Skin temperature	14.91	K
Evaporation	0.005	m of water equivalent
Total precipitation	0.027	m
Leaf area index, high vegetation	6.0	m${}^{2}$·m${}^{-2}$
Leaf area index, low vegetation	4.106	m${}^{2}$·m${}^{-2}$
Volumetric soil water layer 1	0.5	m${}^{3}$·m${}^{-3}$
Volumetric soil water layer 2	0.503	m${}^{3}$·m${}^{-3}$
Volumetric soil water layer 3	0.507	m${}^{3}$·m${}^{-3}$
Volumetric soil water layer 4	0.512	m${}^{3}$·m${}^{-3}$

**Table 2 sensors-23-07632-t002:** Training and test loss using MAE and MSE for the vanilla CNN and CIWA-net models; the number of delayed GRACE data taken as additional input channels (δ) was varied. Bold numbers are the best test scores for the CNN model. Underlined bold numbers are the overall best test scores.

		δ= 0	δ= 1	δ= 2	δ= 3	δ= 4	δ= 5
***vanilla*** **CNN**	*Train MAE*	0.00440	0.00435	0.00328	0.00743	0.00547	0.00669
	*Train MSE*	0.00004	0.00004	0.00002	0.00012	0.00007	0.00011
	*Test MAE*	0.03940	**0.03426**	0.03593	0.03836	0.03585	0.03569
	*Test MSE*	0.00304	**0.00242**	0.00267	0.00278	0.00260	0.00261
**CIWA-net**	*Train MAE*	0.01932	0.01948	0.01258	0.01490	0.02425	0.01639
	*Train MSE*	0.00074	0.00078	0.00033	0.00046	0.00146	0.00055
	*Test MAE*	0.04461	0.03479	** 0.03189 **	0.03273	0.03861	0.03435
	*Test MSE*	0.00407	0.00218	** 0.00192 **	0.00200	0.00281	0.00217

## Data Availability

Images on cocoa pods were retrieved at: https://www.kaggle.com/datasets/serranosebas/enfermedades-cacao-yolov4 (accessed on 19 May 2023); the dataset with the input land and meteorological features from ERA5 was retrieved at: https://cds.climate.copernicus.eu/cdsapp#!/dataset/reanalysis-era5-single-levels-monthly-means?tab=form (accessed on 19 May 2023); the dataset with the target feature “equivalent water thickness” from GRACE was retrieved at: https://www2.csr.utexas.edu/grace/ (accessed on 19 May 2023).

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
