# Peer review of "Machine Learning as a Strategic Tool for Helping Cocoa Farmers in Côte D’Ivoire"

_sensors, 2023, doi:10.3390/s23177632_

Round 1
Reviewer 1 Report
1. There are.many sensors have been presented,so what sensors did this paper used ?
2. Only the images used here by the machine learning, is there any other situation.
3. The open source tool is not clearly presented.
4. The results is simple,more case study details should be added.
5. Conclusions are long, the authors should simply the words and present the highlights of this paper.
6. The discussion and abstracts should be improved further.
Author Response
Please see the attachment file. Your comments appear as reviewer #1.

Reviewer 2 Report
The study is interested.
Not advisable to have citation in the Abstract.
The findings should be reported in the Abstract.
There are many other studies discussed climate change and the use of CNN in their work: Separation and attribution of impacts of changes in land use and climate on hydrological processes, Conceptual Sim-Heuristic optimization algorithm to evaluate the climate impact on reservoir operations, Optimal operation of hydropower reservoirs under climate change
The authors are encouraged to add Flowchart to summarise the methodology section
The structure of the paper is not clear to the reader for example the presenting task per task. Maybe the authors can think in better way in presenting the work. Just suggestion.
There are recent studies about the use of LSTM and CNN in engineering: Graph convolutional network – Long short term memory neural network- multi layer perceptron- Gaussian progress regression model: A new deep learning model for predicting ozone concertation
Figure 6. b): not clear, please improve.
More discussion is needed to be included in the Discussion section.
Author Response
Please see the attachment file. Your comments appear as reviewer #2.

Reviewer 3 Report
Reviewers Comments
Major revision is being suggested for the manuscript id: sensors-2562974, titled, “Machine Learning as a Strategic Tool for Helping Cocoa Farmers in Côte d’Ivoire”. The following are specific comments; the author must revise the manuscript and prepare a rebuttal to the comments for further review.
1. Since this is a research paper, the author must first read the article template provided by sustainability Journal. It must strictly adhere to the IMRaD (Introduction, Materials and methods, Results and discussion, Conclusions, and References) template; the author must restructure the paper to adhere to the template. The link of the template: https://www.mdpi.com/files/word-templates/sensors-template.dot
2. Authors must avoid the use of first-person pronouns (Line no. 1, 8, 9,12,14, 17, 134 etc.) in the entire manuscript start from abstract till the conclusion.
3. In the entire manuscript as well the title, the author must avoid unnecessary capitalization. The entire paper needs to be reviewed and updated.
4. Introduction and related work should be merged. Here in the manuscript no proper formulation of the research gap and the objectives of the presented work. Hence it is suggested that author should structure the entire section such that the research gap is offered first, followed by the objectives of the presented study in the final paragraph of the introduction.
5. The section “Open-Source Strategic Tools” should be shifted in Section 2: Materials and Methods. Als author must discuss first about the open-source strategic tools in brief and present the methodologies adopted for task 1 and task 2.
6. The caption of the table must be above the table. (After line no 374 and 451).
7. The third section should be titled Results and discussion. The results must be presented and interpreted with adequate rationale, which the developed model completely lacks for such behaviour. Also, Task 1 and Task 2 must be restructured and the contents must be readjusted as per the revised section titles.
8. The current discussion part is of no use, here author presented the contents which is non related to the thematic idea of the work. Hence it is suggested to be removed and the obtained results in the task1 and task 2 must be properly presented under earlier section results and discussion by comparing the obtained results with the existing work and try to present the root causes of the improvement of degradation in the quality of the developed models.
9. Current section: Future work, non-related explanation, it can be fully removed.
10. Conclusions must be rewritten to no more than 200 words. It must present the major findings.
I wish authors a great success.

Author Response
Please see the attachment file. Your comments appear as reviewer #3.

Round 2
Reviewer 1 Report
The paper was revised well.
Reviewer 3 Report
Authors have addressed all the comments, hence it can be accepted for publication. However future work must be shifted after the conclusions in the final version of the manuscript.